# Programme for the Effective Promotion of Maternal Psychosocial Wellbeing (PREPWELL) in Ghana: Development and field-testing of a mHealth Intervention in a rural setting

Benedict Weobong[1,2]* Solomon Nyame[3], Dzifa Attah[4], Kenneth Ae-Ngibise[3], Joseph Osafo[5], Betty Kirkwood[6], Angela Ofori-Atta[4], Kwaku Poku Asante[3], Philip Baba Adongo[1]

1 Department of Social and Behavioural Sciences, University of Ghana School of Public Health, Toronto, Ghana, 2 School of Global Health, York University, Accra, Canada, 3 Kintampo Health Research Centre, Ghana, Research and Development Division, Kintampo, Ghana, 4 Department of Psychiatry, University of Ghana Medical School, Accra, Ghana, 5 Department of Psychology, University of Ghana, Accra, 6 Department of Population Health, Faculty of Epidemiology and Population Health, London School of Hygiene and Tropical Medicine, London, United Kingdom

* bweobong@yorku.edu

## Abstract

Poor maternal psychosocial well-being, often manifested as mental health challenges, is a significant public health concern – particularly in settings with limited access to mental health care. Although Behavioural Activation (BA) has shown strong promise in promoting psychological well-being, there is a notable lack of comparable evidence from low- and middle-income countries (LMICs). While the WHO advocates for the innovative delivery of psychosocial interventions via mobile health (mHealth) platforms, supporting data from LMIC contexts remain limited, underscoring the need for context-specific research and implementation. The PRogramme for Effective Promotion of maternal psychosocial WELLbeing (PREPWELL) sought to address this innovation and knowledge gap. Through our three-stage intervention development process – (a) formative research to identify contextual information regarding the feasibility and acceptability of *Obaatanpa*; (b) constructing a prototype *Obaatanpa* by adapting an evidence-based psychological treatment known as the Healthy Activity Programme; and (c) field testing, our research co-created a prototype psychosocial wellbeing promotion mHealth intervention (*Obaatanpa*) for mothers in Kintampo, Ghana. Findings showed a phasic intervention comprising psychoeducation, BA, and problem-solving delivered over 8 sessions through mobile phones is feasible, acceptable, comprehensible, and salient. However, the drop in participation from session 6 suggest a preference for smaller number of sessions but this needs further unpacking in a pilot study. The data also suggests that a core aspect of BA such as doing pleasurable activities – though possibly embedded – were less well developed, indicating an area for possible emphasis or clearer articulation in future iterations. With these

**Data availability statement:** Data availability statement: The data may be accessed from this data repository (https://doi.org/10.71620/khrc.32159643). There are minimal restrictions due to potentially identifying information. Interested parties may contact the Kintampo Health Research Centre through the following email contact to request access: enquiries@kintampo-hrc.org.

**Funding:** This study received funding from the University of Ghana Research Fund (UGRF/12/M-DG-020/2018-2019 to BW). The funders had no role in study design, data collection and analysis, decision to publish, or preparation of the manuscript.

**Competing interests:** The authors have declared that no competing interests exist.

enhancements, and further testing with a larger sample, *Obaatanpa* could be scalable and underscores the potential utility of mHealth as a platform for implementing WHO's self-care framework. The PREPWELL programme of work has demonstrated the feasibility of adapting a treatment-focussed psychosocial intervention for promoting mental health and wellbeing.

## Introduction

Maternal psychosocial well-being is a concept that defines the psychological and social aspects of motherhood. At one of its extremes, this concept encounters maternal perinatal depression, a condition that is now recognized as a major public health issue worldwide. Recent [1] and past [2] reviews show that the prevalence of perinatal depression is significantly higher in low-and middle-income countries (LMICs) – 26.3% during pregnancy and 23.1% postpartum – compared with 11.3% and 18.3%, respectively, in high-income countries (HIC). The consequences are well documented [3–5] and evidence from a cohort study in Ghana, suggests perinatal depression can increase the risk of infant morbidity and mortality [6,7]. The adverse effects on infant survival and development are exacerbated by the mother's poor mental health, yet in Ghana, its detection in routine primary care is almost non-existent [8]. Further, only 2% of persons with a diagnosis of a mental health condition in Ghana get some form of treatment for their mental health condition [9,10]. This highlights the urgent need for bold and innovative strategies – such as preventative, community-based mental health promotion interventions, strongly endorsed by proponents of global mental health – to be integrated within existing reproductive and child health services in LMICs. Despite the promise of such approaches, there remains a significant research gap: only 6% of all trials aimed at preventing the onset of major depression have been conducted in LMICs [11].

This shift in paradigm is supported by the WHO self-care framework that recognises that treatment-oriented approaches alone are not sufficient to tackle the increasing burden of disease [12]. Particularly for mental health, there is good reason to theorise that treatment-oriented approaches such as evidence-based psychological treatments (PTs) can be leveraged for mental health promotion. To illustrate, there is growing evidence to support the idea that well-being can be enhanced through positive psychology interventions (PPIs); PPIs enhance well-being and ameliorate depressive symptoms, and that these effects are enhanced for individuals with depression [13].

One psychological treatment that aligns with the PPI stance is Behavioural Activation (BA). BA is a form of psychological intervention originally developed for the treatment of depression that seeks to identify and promote engagement with activities and contexts that are reinforcing and consistent with an individual's long-term goals [14]. Mazzucchelli and colleagues have noted that although behavioural activation has historically been positioned as a treatment for depression, it differs little in content from some behaviourally focused positive psychology interventions, except in the intent guiding their application [13]. Their meta-analysis on BA interventions for the

promotion of psychological well-being in the general population suggests moderate positive effects (Cohens d = 0.52) with a little over two-thirds of control participants recording improvements less than average effects of BA [13]. Probable mechanisms explaining the role of BA in promoting well-being are drawn from Seligman's proposed components of a happy life through positive emotion, engagement, and meaning [15], and it might be argued that BA targets all of these three components. Further, the behaviours taught in BA – such as increasing activation and reducing rumination – help individuals manage emerging symptoms before they escalate, thereby improving emotional regulation in ways that may also reduce interpersonal conflict, including domestic violence [16–18]. However, despite the very promising data about the potency of BA for the promotion of psychological well-being, there is no reported comparable data in LMICs. Furthermore, to address severe access challenges, WHO recommends the innovative delivery of PTs through mHealth platforms [19], but there is limited data in LMICs to support this. We argue this is because there is a gap in knowledge regarding the methodology for adapting a PT for non-targeted delivery, anchored on mHealth platforms. Digital technology is strongly recommended by the Lancet Commission on Global Mental Health and Development as a tool with enormous potential in bringing changes in mental health care [20]. Mobile penetration in Ghana is now well over 100% [21], and this offers a strong platform for mHealth interventions targeting maternal mental health. Two notable mHealth programs: The Mobile Technology for Community Health (MOTECH) "Mobile Midwife" [22] and Technology for Maternal and Child Health (T4MCH) intervention have both demonstrated that SMS and voice messaging can improve maternal knowledge and care-seeking (antenatal care attendance, skilled delivery, postnatal care use). These initiatives have focused almost entirely on service utilization and physical maternal outcomes; they do not directly address perinatal mental health. A recent study highlights the need for mobile-supported tools for women's mental health in Ghana [23].

PREPWELL sought to address this innovation and knowledge gap by embarking on a phased programme of work to offer mothers the opportunity to acquire essential skills to take care of their own mental health during particularly heightened periods of vulnerability such as pregnancy and after birth. As reported in the recent World Mental Health report (chapter 6, page 171) we agree with Olivia's admonishing that every woman going into motherhood should be educated on how to take care of their mental health [24].

In this paper, we report the first phase involving the systematic development of a co-created prototype psychosocial well-being promotion intervention *Obaatanpa (good mother)* for mothers in Kintampo, Ghana. The development process as described was guided by the following objectives: 1) to adapt a BA-based psychological treatment – Healthy Activity Program (HAP) into a culturally acceptable guided self-help psychosocial well-being intervention for delivery to pregnant women and mothers who have recently given birth in the Kintampo North Municipality of Ghana; 2) to assess *Obaatanpa*'s feasibility, comprehensibility, acceptability, and saliency for pregnant women and women who have recently delivered in the Kintampo North Municipality of Ghana. This would contribute important data towards an evaluation phase involving a feasibility trial and a subsequent definitive randomised controlled trial to evaluate effectiveness outcomes.

## Materials and methods

### Ethics statement

All research procedures with participants contained in the research protocol were approved by the University of Ghana College of Health Sciences Ethical and Protocol Review Committee (CHS-Et/M3-7.16/2019–2020), and the Kintampo Health Research Centre Institutional Review Board (KHRCIEC/2020-03). Written informed consent was obtained from participants.

The intervention development involved three main stages through multiple steps of research (a) formative research to identify key contextual information regarding the feasibility and acceptability of the *Obaatanpa* intervention; (b) constructing a prototype *Obaatanpa* by adapting an evidence-based psychological treatment known as the Healthy Activity Programme (HAP) to identify specific components that are feasible and acceptable to mothers and other stakeholders;

and (c) field testing the prototype intervention. Fig 1 summarises the procedures used in each stage. The details of each stage are discussed below. Broadly, we conducted a situation analysis, intervention adaptation, and field-testing. A variety of methods were used including desk review, elicitation of factual information provided through key informant interviews (KIIs) and participatory workshops with mothers (pregnant/recently delivered), husbands, health administrators, and health workers and naturalistic observation at health facilities. A multi-stage short messaging system (SMS) content development methodology was also employed. The fieldwork was carried out in the Kintampo North municipal area of the Bono East Region of Ghana. The SMS system was developed by Viamo [25], a local technology partner for this project. Viamo is a social enterprise specializing in using mobile technology to innovate health solutions. Viamo's connections allow access to call every phone on the planet, and its network of local infrastructure allows it to simultaneously push and receive content at high volumes. The study took place between March 31st, 2020, and November 30th, 2022.

### Stage 1: Identifying key contextual information on maternal mental health services

The goal of this stage was to obtain information on several relevant factors including: digital health interventions for mothers, mental health services for mothers within the study area, cultural appropriateness of the concept of Guided Self-Help, and feasibility of the PREPWELL programme. This stage involved steps A and B.

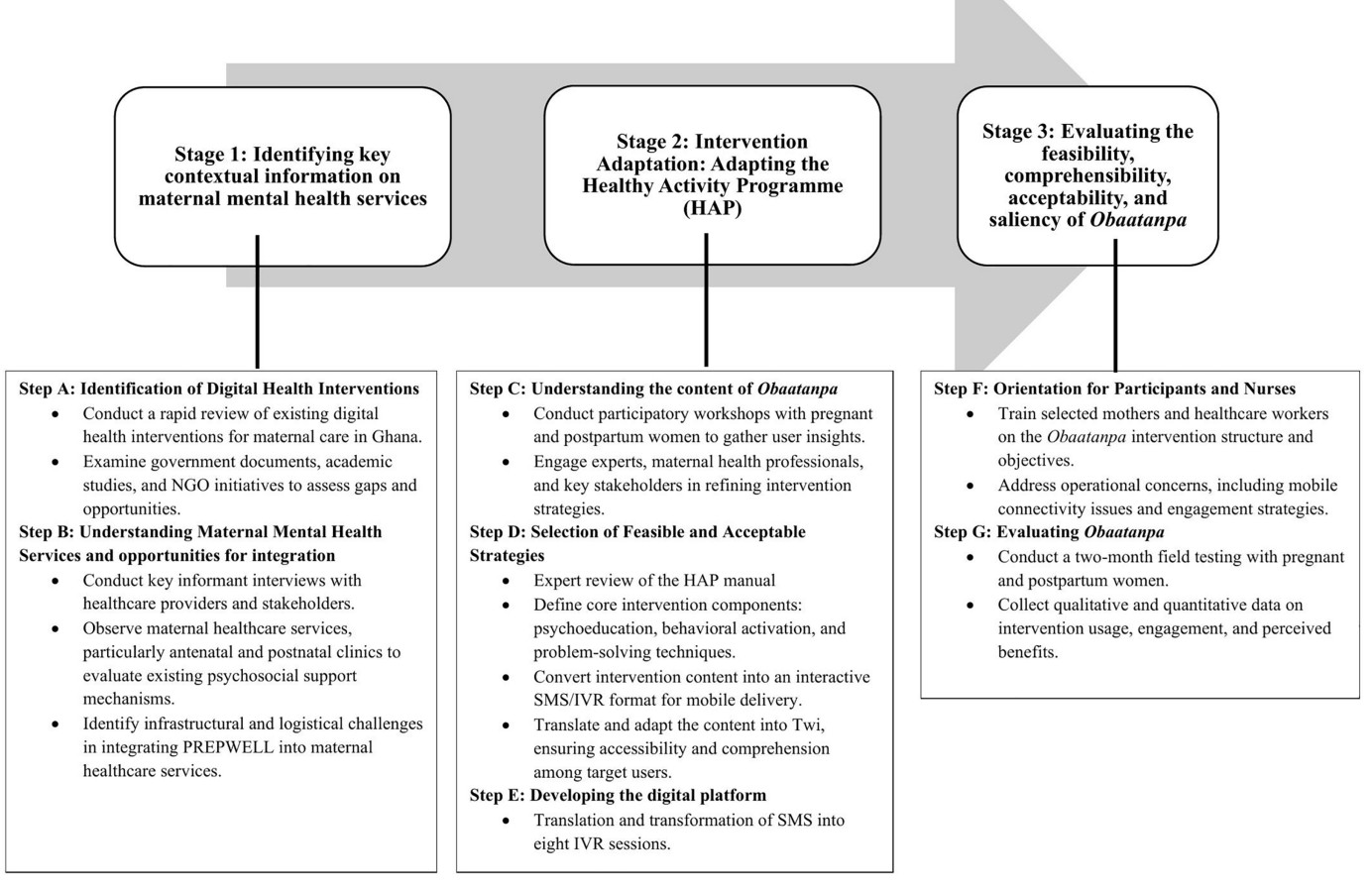

**Fig 1. Overview of the development process of *Obaatanpa*.**

**Step A: Identification of digital health interventions for mothers.** Our formative research began with a rapid review on digital health interventions for mothers in Ghana, and within the specific study area. The goal was to learn what already exists and opportunities for further development. We opted for the rapid review approach to narrow the scope of the research question and reduce the extent of data abstraction to provide evidence in a timely and cost-effective manner particularly because we had limited resources to conduct a full systematic review [26]. The review involved examination of official government documents, online searches of governmental and non-governmental initiatives, and academic studies/reports (we searched PubMed, Medline, PsycINFO, Websites of Government of Ghana, relevant Ministries, Departments and Agencies, Scopus, Google Scholar). The search strategy for electronic databases incorporated both medical subject headings (MeSH terms) and free-text key terms adapted to suit each individual database using applicable controlled vocabulary. The key search terms used in this review are ((("maternal health"[All Fields] OR ("mother"[MeSH Terms] OR ("pregnancy"[All Fields] OR ("antenatal"[All Fields] OR ("antepartum"[All Fields] OR ("postpartum"[All Fields] OR ("postnatal"[All Fields] AND "mHealth"[All Fields]) OR "digital health"[All Fields])) OR "mobile health"[All Fields])) OR "ehealth"[All Fields])) OR "USSD"[All Fields])) OR "SMS"[All Fields])) AND ("ghana"[-MeSH Terms] OR "kintampo"[All Fields]). We included studies reported between 2010 and 2020.

**Step B: Understanding maternal mental health service delivery and opportunities for integration.** Two approaches were also used in this step: (a) semi-structured interviews to collect information on: policies on maternal mental health, availability of mental health/psychosocial health service and referral systems; acceptability of a mobile phone delivered mental health intervention. Interviews were conducted with health directors, public health nurses, midwives, community psychiatric nurses, clinical psychologists, community mental health officers, community psychiatric officers, community health planning services coordinators, disease control officers; and (b) naturalistic observations at one antenatal care (ANC) and one postnatal care clinics to document the natural flow of activities in terms of: numbers attending, pregnancy clinics, schedule of activities, waiting times, mobile phone penetration, including assessment of engagement and enrolment procedures at the antenatal clinics.

### Stage 2: Intervention adaptation: Adapting the Healthy Activity Programme (HAP)

The goal of this stage was to construct the design and features of the *Obaatanpa* intervention. This involved co-producing the content in terms of the mental health conditions to include in the intervention, the HAP components to maintain, target mental health conditions for *Obaatanpa* and acceptability enablers. This stage involved steps C, D, and E.

**Healthy Activity Programme (HAP).** The theoretical foundation for *Obaatanpa* is Behavioural Activation (BA) Therapy [27], an empirically supported psychological treatment recommended by the WHO [28]. This is the underpinning theory behind the Healthy Activity Program (HAP) that we adapted for *Obaatanpa.* HAP is a brief evidence-based psychological treatment [17] that is gaining popularity in India, Nepal [29] and very recently Uganda [30]. HAP is delivered face-to-face by a trained lay counsellor over 6–8 weekly sessions and focuses on increasing patient activation levels in pleasurable or mastery activities, and comprises the following strategies: psychoeducation, behavioural assessment, activity monitoring, activity structuring and scheduling, activation of social networks, problem-solving, and assertion training. HAP was developed and evaluated for the treatment of moderate to severe depression among adults attending primary health care in India using a parallel-arm, individually randomised controlled trial which found an effect size of 0.48 and a 64% remission rate at 3 months [17] which were sustained at 12 months [18].

**Step C: Understanding the content of *Obaatanpa*.** Two approaches were used in this step to co-produce the content of *Obaatanpa*: (a) a series of three participatory structured workshops with only pregnant women, only women who have recently delivered and a mix of these two groups. The workshops were conducted in three communities purposively selected for ease of access and presence of a community health nurse. The workshops employed participatory learning and action (PLA) techniques such as free listing, ranking [31] (where participants order choices/needs based on importance) and visualisation (where participants express their ideas in graphical forms) in participatory processes [32],

to actively engage the participants and make the workshops exciting. For example, structured workshop guides were used to guide the completion of five key activities: experience of health-seeking; willingness to receive counselling; mental health conditions of importance; psychosocial well-being areas of importance; and preference for HAP elements. A final consensus-building workshop with experts and practitioners in maternal and mental health, mental health user groups, and Ghana Health Service stakeholders to confirm and validate the emerging content and mode of delivery of the intervention. The guides (available as S1 Text) were developed by the research team to capture the objective of each of the five activities. For example, the guide used to elicit preferences for psychosocial well-being was developed based on Ryff's six domains of psychological well-being [33]. The guide used to identify participants' preferred HAP strategies for inclusion in the intervention was informed by the HAP manual.

**Step D: Selection of feasible and acceptable strategies.** Multiple activities were conducted in this step. Firstly, based on the findings from step D, an experienced clinical psychologist was engaged to review the HAP manual with the aim to select the most appropriate and feasible strategies that can promote mental health and wellbeing. Secondly, the clinical psychologist together with the PI agreed on the most appropriate design (e.g., phases and sessions) and features (e.g., delivery approach) of the evolving digital intervention. Following this, the clinical psychologist compiled an intervention manual detailing the phases of the intervention and the content of each session. The content of the manual was converted to SMS in line with Odeny's SMS content development framework [34] that guided the entire process. This activity was completed by the PI in close collaboration with the technology partner who provided the template for creating SMS-type message trees and logic rules.

**Step E: Developing the digital platform.** The technology partner translated the English SMS into Twi (the most widely spoken local language in the study area) and transformed these into interactive voice response (IVR) content (in the Twi language). The translation process was handled by the technology partner using experienced bi-lingual experts. The research team reviewed the recorded translations for accuracy and completeness before these were programmed for deployment. It had 8 sessions which were deployed on a weekly basis according to the preferred day and time of the week. The platform was configured to implement three retry attempts, spaced five minutes apart, for scheduled calls. An inbound line was set where mothers were able to call in anytime to continue from where they left off or take the sessions if they ever missed a call. An SMS alert was sent to project nurses to follow-up with mothers who reported poor mood in a given session. *Obaatanpa* was administered from Viamo's platform for one month (between 2nd September 2022 and 27th October 2022).

## Stage 3: Evaluating the feasibility, comprehensibility, acceptability, and saliency of *Obaatanpa*

The goal of this final stage was to systematically evaluate the comprehensibility of the PREPWELL approach and content, and its feasibility, acceptability, and saliency using all elements of *Obaatanpa*. This stage involved steps F and G.

**Step F: Orientation for participants and nurses.** In this step, eligible mothers who provided informed consent to be included in the field-testing and nurses (n = 3) were taken through the PREPWELL programme and what to expect with the *Obaatanpa* digital intervention. Prior to this, the technology partner Viamo provided a catalogue of operational considerations regarding the *Obaatanpa* mobile phone-based intervention. These were discussed with the mothers and nurses, and concerns addressed. For example, mothers were informed that they would not incur any charges for answering automated calls or interacting with the *Obaatanpa* mobile phone–based intervention. They were advised that *Obaatanpa* would initiate scheduled calls, with up to three call attempts made at five-minute intervals. Each call constituted a session lasting approximately five minutes. Mothers were also informed of an inbound phone line that allowed them to call in at any time to resume a session from where they had left off or to complete sessions they had missed. Finally, mothers were informed that if they were unable to complete two consecutive sessions, calls from *Obaatanpa* would be discontinued.

**Step G: Evaluating *Obaatanpa*.** The prototype *Obaatanpa* mobile phone-based intervention was evaluated using a field-testing approach with a small group (8 pregnant; 8 recently delivered mothers) of mothers attending antenatal and

postnatal care clinics in one health facility in the study area. Aside from pregnancy and postnatal status, other eligibility criteria included: access to a mobile phone and being able to understand the Twi language. Primary data sources for the evaluation included the 'app' usage survey from the technology partner and session experience data collected using a semi-structured form after the end of each session, and a focus group discussion (FGD) at the end of the field-testing phase. The outcomes of interest were: comprehensibility (how understandable the sessions/messages are), feasibility (recruitment, follow-up, delivering *Obaatanpa* over mobile phone), acceptability (session uptake, engagement, safety), and saliency (how relevant, useful and applicable the HAP skills are to mothers). Assessments using the locally validated Twi version of the Patient Health Questionnaire-(PHQ-9) [35] before and after receiving *Obaatanpa* were interviewer-administered to check safety (no severe symptom deterioration). The PHQ-9 was also administered at the start of each session.

## Data analysis

Framework analysis [36] was used to analyse the main qualitative data from the IDIs conducted with the 16 participants who were involved in the field-testing to evaluate the acceptability, feasibility, comprehensibility, saliency, and safety. In the process of the framework analysis, we manually used the five stages of framework analysis outlined by Ritchie and Spencer [37]: familiarization, identifying a framework, indexing, charting, mapping and interpretation. For the structured participatory workshops, most of the data were quantitative in nature. These were tallied and ranked. Furthermore, the intervention usage data were collected through a built-in survey on the digital platform. All quantitative data were analysed using simple descriptive statistics to summarize information using counts, means and proportions.

## Results

### Participants

A total of 59 participants were involved in various aspects of the study: key informants (n = 16); workshops (n = 27 women); field testing (n = 8 pregnant women; 8 recently delivered). Table 1 shows the characteristics of the mothers who participated in the field-testing of *Obaatanpa.* Two-thirds were young adults, and more than half were married. Most of the mothers had received some form of formal education, though two-thirds reported difficulty in reading.

### Stage 1: Key contextual information identified

**Technology-enabled health interventions.** The rapid review found evidence of technology enabled health interventions targeting mothers in Ghana. The Mobile Technology for Community Health in Ghana MOTECH [38] is a good example worth discussing. MOTECH consists of "Client Data Application" which allows providers to digitize and track service delivery information for women and infants and "Mobile Midwife" which sends automated educational voice messages to the mobile phones of pregnant and postpartum women. The technology was evaluated to assess the platform's effectiveness in delivering messages along with user response across sites in five districts from 2011 to 2014. The authors concluded that the adoption of MOTECH to improve maternal, newborn, and child health service delivery and uptake represents good value for money in Ghana and should be considered for expansion. Other mHealth interventions were identified. A pilot study evaluated healthcare professionals' perceptions of the benefits and challenges of teleconsultation services involving community health nurses [39]. The study concluded that teleconsultation services are potentially important for improving health care services in rural communities.

A synthesis of these studies suggests important learnings for the uptake of technology enabled public health interventions: a) it is important to consider mobile phone ownership dynamics; b) stakeholder involvement from the start is critical; c) technical profile of the initiative should maintain values of simplicity, interoperability and adaptability; and d) ensure privacy and confidentiality.

**Table 1. Characteristics of field-testing participants.**

| Description | Participants (N = 16) n (%) |
|---|---|
| **Perinatal period** | |
| Antenatal | 8 (50) |
| Postnatal | 8 (50) |
| **Age-group** | |
| 20 – 29 years | 10 (62.5) |
| 30 – 39 years | 6 (37.5) |
| **Marital Status** | |
| Married | 9 (56.3) |
| Living together | 7 (43.7) |
| **Educational level** | |
| None | 4 (25.0) |
| Primary | 7 (43.7) |
| Junior High School | 2 (12.5) |
| Senior High School | 3 (18.8) |
| **Ability to read** | |
| Easy | 2 (12.5) |
| With difficulty | 10 (62.5) |
| Not at all | 4 (25.0) |
| **Religion** | |
| Catholic | 8 (50.0) |
| Protestant | 1 (6.2) |
| Pentecostal | 2 (12.5) |
| Muslim | 5 (31.3) |
| **Occupation** | |
| Trader | 4 (25.0) |
| Farmer | 12 (75.0) |

**Maternal mental health service delivery and opportunities. ANC environment:** pregnant women attending antenatal clinics are attended to by health staff including midwives, nurses, and nurse assistants. There is a waiting area for pregnant women. Consecutive pregnant women are invited, and their vital signs, weight measurements, and urine tests are conducted. The services provided also include physical examination and health education on pregnancy care. Routine mental health assessments are not conducted.

**Mental health/psychosocial health services for mothers:** data from interviews with key stakeholders showed that, while there are no dedicated counselling services for mothers within antenatal clinics, a psychosocial support unit linked to the main municipal hospital, run by an academic training institution within the study area, provides support. Despite the existence of this psychosocial support unit, when mothers were asked during the participatory workshops if they have been to see a health worker within the past 3 years on how to take care of their own mental health during pregnancy and after delivery or for a mental health problem in general, a little over 90% reported no such experience. The mothers were however willing (over 90%) to receive counselling if this was available. To buttress this call and willingness for counselling services, the mothers shared a troubling incident where a woman who had recently given birth almost killed her baby because of her poor mental health – in their own words *'it really troubled and disturbed the family so, she thinks it is important to seek help'*

**Acceptability of mobile phone delivered mental health promotion intervention:** we explored what mothers thought about receiving the intervention on their mobile phones and what would make it attractive. Participants thought it was a good idea but were clear that it should not involve use of text messages but should be voice/audio messages in the local language (Twi). They also were clear that the messages should be comprehensible ('easy to understand'), and the intervention should not bring a financial burden ('the intervention should be free').

**Mental health conditions of importance:** we also invited mothers to identify mental health conditions that are important to them and what the intervention should focus on. Fig 2 shows mothers across the three groups wanted help with dealing with depression, followed by stress and anxiety.

## Stage 2: Potential *Obaatanpa* elements identified

**Psychosocial and wellbeing areas:** given that PREPWELL has adopted a psychosocial wellbeing promotion stance, we asked mothers to select from Ryff"s six psychosocial wellbeing domains [33] they would want the intervention to address. Fig 3 shows that taken together, most mothers placed emphasis on being able to make decisions on their own (autonomy). Being able to feel in charge of their living situation (environmental mastery) was also highly rated. These were followed by a keen interest in gaining skills on how to maintain positive relations with others (relationships); and how to remain focussed and not wander aimlessly through life (purpose in life).

**HAP strategies:** The theoretical foundation for *Obaatanpa* is Behavioural Activation (BA) Therapy. This is the underpinning theory behind the Healthy Activity Program (HAP) that we adapted for *Obaatanpa.* We thus asked the mothers to decide the HAP strategies that they would want to see in *Obaatanpa* to help address the psychosocial wellbeing areas that were previously identified as important. Fig 4 shows the strategies mothers wanted most were: understanding HAP and how to apply it to their daily life, doing pleasurable activities, identifying causes and solutions to poor mental health/ low mood/poor psychosocial well-being (learning together), and problem-solving/dealing with barriers.

**Emerging *Obaatanpa* intervention framework (content, features, design).** By triangulating data from stages 1 and 2, the evolving intervention framework for promoting maternal psychosocial wellbeing is shown in Table 2. The core content of the intervention consists of psychoeducation, behaviour activation, and problem-solving. The approach is collaborative,

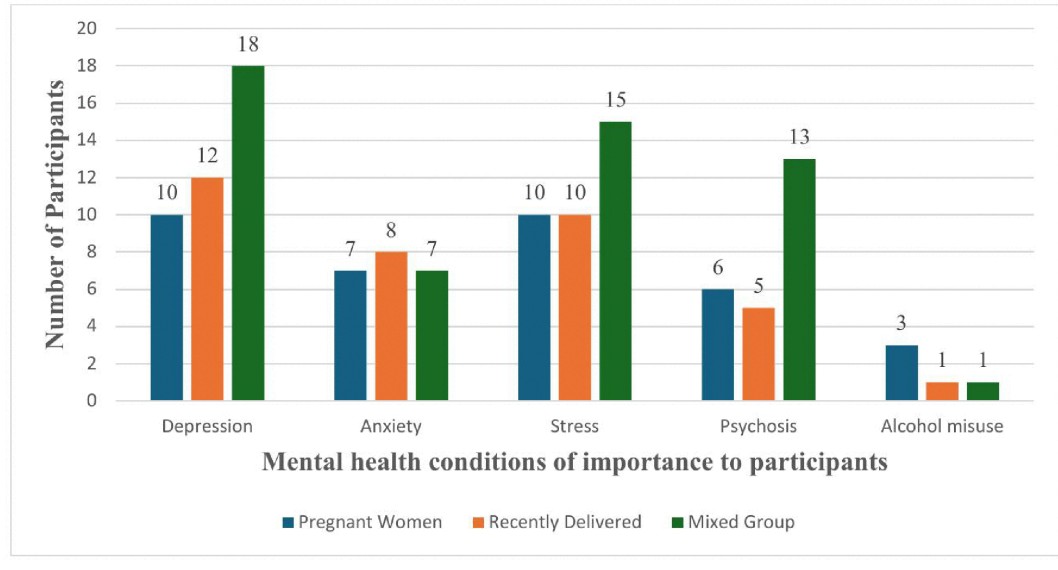

**Fig 2. Mental health conditions of importance to participants.**

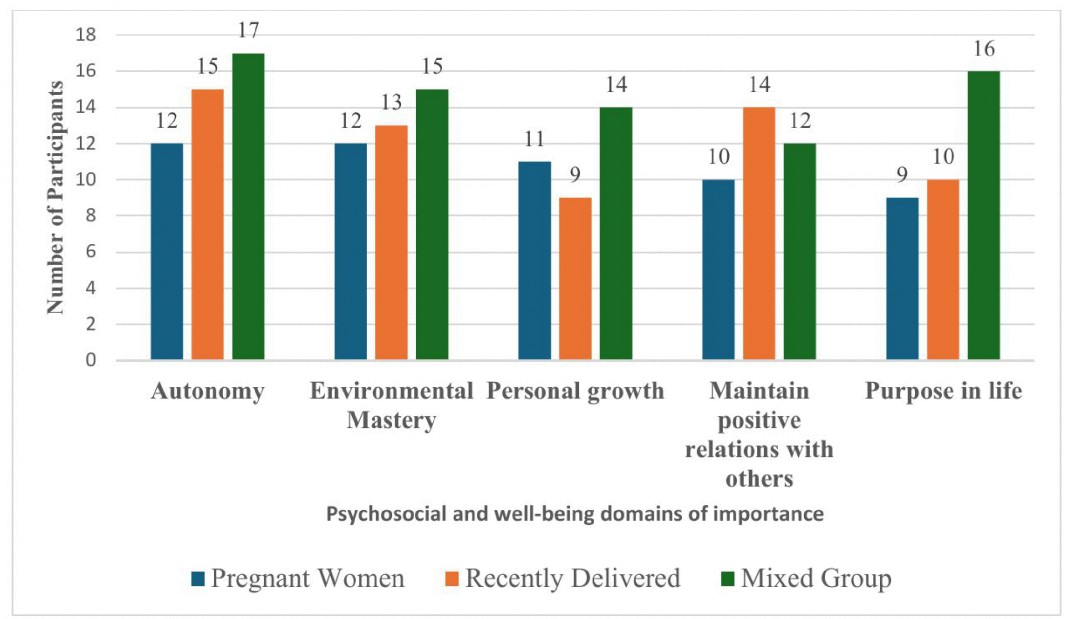

**Fig 3. Distribution of psychosocial and wellbeing domains of importance.**

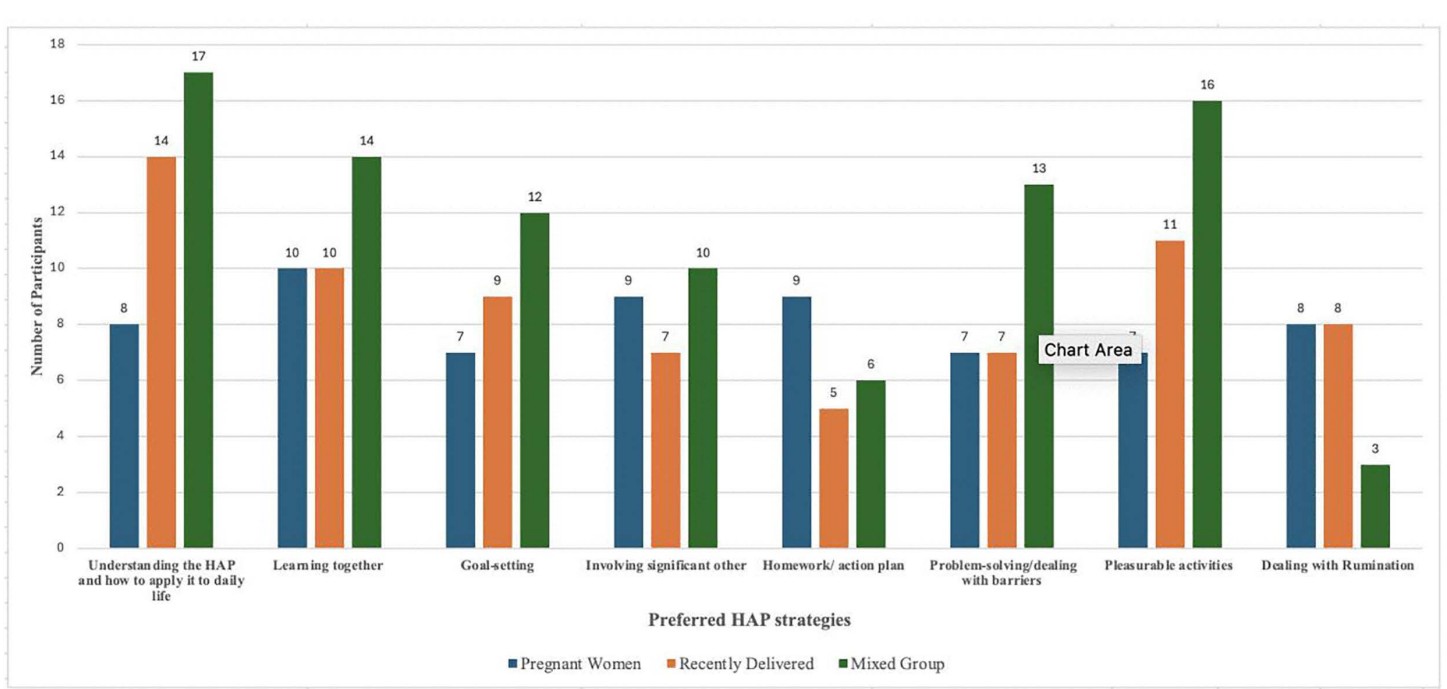

**Fig 4. Distribution of preferred HAP strategies.**

**Table 2. Evolving *Obaatanpa* intervention framework.**

| *Obaatanpa* core content | Featured HAP strategies/ingredients | *Obaatanpa* strategy | Target psychosocial wellbeing domain |
|---|---|---|---|
| Psychoeducation on Stress, Depression & Anxiety | Psychoeducation: explains depression, symptoms, misconceptions, normalises mental health issues. Behaviour–Mood Awareness Cognitive Awareness: link behaviour to mood, reinforce positive cycles | Culturally tailored stories (*Obaatanpa*, Ama, Naa, Abena); emphasis on pregnancy/postpartum risk factors; destigmatising maternal distress. Stories and guided reflections show how actions influence emotions; emphasise maternal empowerment ("what you do affects how you feel") | Self-acceptance Personal growth Purpose in Life |
| Behaviour Activation: increases engagement in pleasant activities activity monitoring and scheduling | Behavioural assessment increased engagement in pleasant activities activity monitoring and scheduling | Mothers identify pregnancy-friendly, mood-enhancing activities (e.g., rest, social visits, bonding with baby); tracking via phone calls. | Environmental mastery; Purpose in life; Personal growth; Positive relations with others; Autonomy |
| Problem solving Skills for Maternal Challenges | Structured solving problems problem-solving steps (define, brainstorm, choose, test, review) | Apply problem solving steps to maternal-specific issues (childcare, partner conflict, financial stress, infant care barriers); *Obaatanpa* helps break tasks into smaller steps | Environmental mastery; autonomy; Personal growth; Purpose in Life |
| Mobilising Social Support | Involving significant others; encouraging family and community support | Invites husband, friend, mother, or nurse to support; promotes peer-to-peer support among mothers; integrates Community Health Nurses | Positive Relations with Others; Self-Acceptance; Autonomy |
| Homework & Self-Monitoring (Weekly Exercises) | HAP homework: activity calendars, pleasant activity practice, PHQ-9 feedback | Weekly assignments adapted to maternal context: identifying stress triggers, self-care tasks, involving family in symptom tracking | Autonomy; Environmental Mastery; Personal Growth |
| Relapse Prevention and Maintenance Planning | Ending Phase: review gains, plan for high-risk situations, sustain routines | Final session dedicated to planning for postpartum stressors, reinforcing skills, and maintaining routines beyond programme completion | Purpose in Life; Environmental Mastery; Self-Acceptance |
| Therapeutic Style (Warmth, Encouragement, Cultural Respect) | HAP counsellor style: empathy, non-judgmental support, collaborative agenda-setting | *Obaatanpa* facilitators use Ghanaian proverbs, respectful maternal language, and culturally familiar metaphors to build trust and rapport | Positive Relations with Others; Self-Acceptance |
| Safety Planning/ Crisis Management | HAP safety steps: suicide risk assessment, immediate management | Integrated referral pathways for postpartum depression, domestic violence, and suicide risk; emphasises maternal and infant safety | Environmental Mastery; Self-Acceptance |

empowering, non-judgemental, and adaptable/flexible. The main delivery feature for the content is IVR, including the use of story-telling. The design is an intervention delivered over 8 sessions in 3 phases (Introduction & orientation; Skill building; Maintenance), is interactive, allows for personalised action plans, and requires minimal human contact.

**Stage 3: Acceptability, feasibility, comprehensibility, saliency, and safety of *Obaatanpa***

**Acceptability and safety.** Fig 5 shows the session completion rate by participants; one participant was deemed to have dropped out after failing to respond over two consecutive sessions (drop-out). Subsequent calls from session 3 were directed at the remaining 15 mothers. By session 5, two more drop-outs were recorded and the calls were directed at the remaining 13 mothers. The 13 mothers remained active until session 8 when we recorded 2 drop-outs. The mean session duration was 7.3 minutes (95% CI, 6.3–8.3). Eight (50%) mothers (split equally between pregnant and recently delivered) completed all eight sessions. Between sessions 1 and 5, there was an almost 100% completion at each session, demonstrating a high level of acceptability and usage.

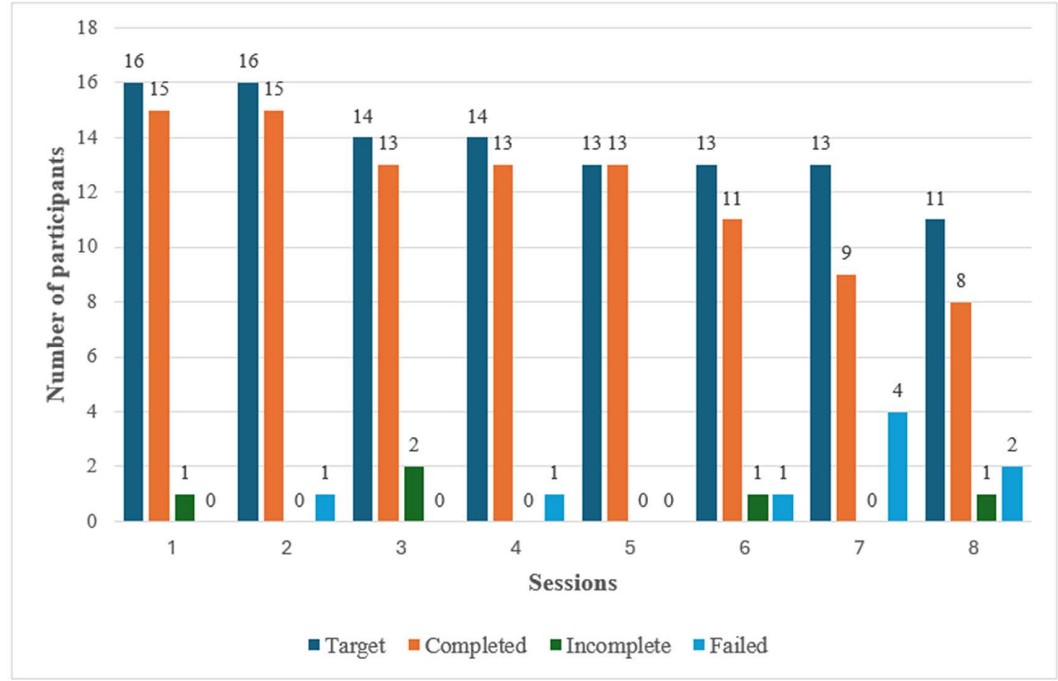

**Fig 5. Session completion among participants.**

In terms of safety, there was no indication of symptom deterioration based on mean PHQ-9 total score at each session 0.39 [SD = 0.34].

*Obaatanpa* was acceptable to the participants. Participants expressed a positive attitude towards the intervention and reported satisfaction with the programme's content and delivery methods. Some participants expressed the following sentiments during FGDs to confirm their satisfaction with the *Obaatanpa:* The quotes below provide some support to this.

*"At first, I used to be worried but now after learning these things from the Obaatanpa, I have become active. I won't worry about certain things again". [A 38-year-old postnatal woman]*

*"I always thank Obaatanpa for these words of encouragement and I urge them to continue" [36-year-old postnatal woman].*

It appears the use of the story-telling approach in delivering the intervention was an important ingredient for the acceptability of *Obaatanpa.* Whilst acknowledging the messages were long, one of the mothers thought this was not an issue because of *Obaatanpa's* account of her own story, which they found to be interesting and relatable.

*"It was long but I didn't feel it because the conversation was interesting" [A 36-year-old postnatal woman]*

Notwithstanding, the outcome of the birth presented a possible barrier to acceptability. One woman shared a poignant experience that affected her participation in the PREPWELL intervention. She explained:

*I stopped because I did not get my baby. So, I have to stop the participation. [A 23-year-old postpartum woman]*

**Feasibility.** Based on our learnings from the situation analysis, we collaborated with the health directorate to select the health facility for the field-testing, and for permission to access the facility. Once at the facility, mothers attending antenatal/postnatal clinics were recruited for the study after they went through scheduled activities.

Aside from demonstrating the feasibility of recruiting mothers from antenatal/postnatal clinics, the delivery of *Obaatanpa* as intended through mobile phones was found to be feasible. The messages were transmitted and delivered as planned. Usage data from the Viamo platform shows that all messages were listened to in all 8 sessions. Whilst the majority (11/15) of the mothers knew how to use phones, some experienced challenges interacting with *Obaatanpa,* particularly in navigating between sessions by pressing buttons on the phone. However, after a couple of tries and with some help from the supervising nurses, participants were able to interact with *Obaatanpa.*

*"For the initial few weeks, it was difficult for me to select the numbers while on the call, your worker showed me how to do it and I was able to do it going forward"*

*[37-year-old postnatal woman]*

The data also suggests that the ability to keep the phone close by and ready to respond to the call from *Obaatanpa* was the feasibility factor, and not access to a phone – majority of the mothers (10/15) had no issue accessing a phone when we enquired about this during intervention delivery. For example, the mothers reported missing the call because the phone was being recharged or attending to other activities. Some mentioned that their phones were charging, which prevented them from participating in scheduled calls or receiving timely communications. These experiences and instances highlight the logistical challenges that can arise in delivering remote interventions and the need for flexible and accessible communication strategies. The following experiences were shared by participants during the FGD session:

*"I was charging my phone, so I did not see the call and because of that we were not able to interact". [A 26-year-old postnatal woman]*

*"I left my phone at home because I was charging it. I went to the market so when the call came through, I was not around to answer the call". [A 25-year-old pregnant woman]*

Additionally, even though the mothers chose the most preferred timings to interact with *Obaatanpa,* there were still some challenges keeping to these preferred timings. One participant noted the following experience.

*"Yes, just that I was feeling sleepy because she called me in the evening". [A 23-year-old pregnant woman]*

This experience highlights the importance of considering optimal timing for engagement to ensure participants are fully attentive and can benefit from the sessions.

**Comprehensibility.** The mothers comprehended and understood *Obaatana,* particularly its purpose, messages on self-care during and after pregnancy, and its emphasis on emotional well-being and resilience. Most mothers mentioned advice received, examples shared, and actionable insights. For example, 8 out of 13 mothers at session 3 were able to identify that what they do affects how they feel, illustrating some understanding of the connection between actions and feelings. Participants shared the following experiences.

*"My understanding was that some women easily get angry during pregnancy and so we must know how to relate with people" [36-year-old pregnant woman]*

*"I enjoy listening to her since the message were clear and easy to understand" [29-year-old postnatal woman]*

> *"She used herself as an example that helped me a lot and made me believe whatever she asked me to do was good."* [A 25-year-old pregnant woman]

Language barriers appeared to affect understanding, but only for a few of the mothers who were probably not native speakers of the Twi language.

**Saliency.** Most of the mothers found the content of *Obaatanpa* meaningful, motivational, and emotionally engaging. The mothers repeatedly described how the intervention encouraged them and helped them cope. They also mentioned how the intervention prompted reflection on their behaviour or mindset and this helped them to know how to address their challenges. The story of *Obaatanpa,* using 'her' own life experiences ensured real-life relevance to the mothers. The mothers found the following suggestions and advice very beneficial: planning ahead, having a goal, and spending time with friends and relatives. Overall, the mothers identified the programme content as relevant to their experiences and perceived it as valuable in promoting their mental and emotional health. Some of these were reflected in the following sentiments.

> *"She [Obaatanpa] said when we are worried over issues, we should try and talk to friends.*
>
> *It helped me to stop some fears and thinking associated with pregnancy and childbirth".* [A 28-year-old pregnant woman].
>
> *"It made me understand that in life we should at least do something and not sit idle."* [A 23-year-old pregnant woman]
>
> *"I need to plan and not think too much."* [A 24-year-old postnatal woman]
>
> *"I was receiving encouraging messages from Obaatanpa which really helped me to come out of the sorrow I was going through at the time".* [A 27-year-old postnatal woman]
>
> *"She used herself as an example that helped me a lot and made me believe whatever she asked me to do was good."* [A 25-year-old pregnant woman]

However, some mothers who dropped out (n = 3/16) admitted that they had not understood what the intervention was about due to language barrier, and also because they lost the baby within the period.

## Discussion

We set out to co-create a prototype psychosocial wellbeing promotion mHealth intervention (*Obaatanpa*) for mothers in Kintampo, Ghana, and evaluate its feasibility, comprehensibility, acceptability, and saliency. Our findings show the initial promise of a feasible and acceptable technology-delivered self-help intervention that seeks to empower mothers to apply simple, evidence-based strategies to help improve and maintain their mental health during periods of heightened vulnerability. With advances in technology, *Obaatanpa* is designed to remain potentially relevant into the future with the flexibility of updating the content of the intervention that can be accessed from the lowest range of mobile phone functionality.

The WHO is promoting self-care as a key strategy for health promotion. Global mental health is prioritizing prevention as a key and strategic approach to tackling the huge disparities in access to mental health services and support in LMICs[20]. Technology-supported mental health interventions are particularly ideal for promoting access through the process of 'digital task-shifting' – a variant of the well-tested human-delivered task-shifting model. Mobile phone ownership in low- and middle-income countries is increasingly widespread [40] offering strategic opportunities to expand access to mental health promotion and support [41]. The philosophy of the PREPWELL programme, embodied through the co-creation of the *Obaatanpa* intervention, directly aligns with and responds to this emerging perspective and growing momentum.

PREPWELL is adding to these growing initiatives and support for the use of mHealth for promoting self-help [20], taking a slightly unique approach. Our positive psychology inspired preventative intervention – *Obaatanpa* evolved from an evidence-based psychosocial intervention designed specifically for the treatment of depression. We have demonstrated the feasibility of adapting a treatment-oriented psychosocial intervention such as the Healthy Activity Program (HAP) for non-targeted delivery. The theoretical stance/orientation of the HAP makes it suitable for remodeling for universal delivery.

Important lessons emerged from this preliminary stage in the development of the *Obaatanpa* intervention. First, conducting a situation analysis to understand the context was important in gauging potential acceptability and feasibility of the intervention. PREPWELL is situated within the domain of mental health promotion. The process of enabling mothers to take care of their mental health and wellbeing needs during pregnancy and after birth may benefit from encouraging mothers to discuss their challenges with health professionals. In the Kintampo setting where the study was conducted, this concept appears alien as 90% of the mothers had not accessed psychosocial support services despite the existence of this service in the district hospital. Our data also showed that these same mothers would welcome the opportunity to engage in counselling if this was offered. This included the option of receiving this support through mobile phone but with the caveat that it should not be text messages. The challenges with use of SMS in low literacy settings has been previously reported [23,42]. Second, for a complex intervention it is essential that the power to decide the content rests with the intended beneficiaries. This approach is enhanced by breaking down complex concepts and adopting less complicated research procedures. We relied heavily on participatory learning and action to elicit components of psychosocial wellbeing and the core strategies of the Healthy Activity Programme to maintain for PREPWELL. This resulted in generating a list of four psychosocial well-being areas that we believe the mothers felt were important to them. The mothers were keen on gaining skills on how to remain focussed on what makes life important, maintain positive relations with others, how to have a fulfilling life, and being able to cope well with overwhelming situations in life. These are meaningful choices, particularly for pregnant and women who have recently delivered. The period could potentially be lonely and overwhelming particularly for mothers in relatively economically disadvantaged regions who maintain many other duties including working and childcare. There is growing momentum in learning essential problem-solving skills for navigating social support, a key determinant of mental health and well-being, as evidenced by existing literature [43]. We enhanced the learning and skills acquisition considering local and culturally relevant strategies. Story-telling was a key strategy employed, and this improved comprehension and relatability. Recent systematic reviews have found that changing health-promoting behaviours using storytelling (i.e., sharing stories) appears to be promising [44,45]. Similarly, the mothers found HAP as attractive and useful in enabling the acquisition and maintenance of the necessary skills to look after their own mental health. The behaviours taught in HAP, such as enhancing activation and identifying and setting goals are core building blocks of psychosocial wellbeing. Probable mechanisms explaining the role of HAP in promoting well-being are drawn from Seligman's proposed components of a happy life such as positive emotion, engagement, and meaning [15], and it might be argued that HAP targets all of these three components [13]. Third, targeted psychosocial interventions have cross-cultural and theoretical applicability. For example, HAP developed in south Asia has shown promise as a relevant theoretical framework for prevention in Africa. There is also evidence that behavioural activation (BA) interventions have moderate positive effects (Cohens d = 0.52) in the promotion of psychological well-being [13]. Fourth, whilst the concept of self-help was confirmed in our setting as arguably alien – 9 out of 10 mothers had not taken the initiative to seek help in the past 3 years on how to take care of their own mental health during pregnancy and after delivery, they were positive and willing to access such opportunities.

Nevertheless, some limitations are worth reflecting on in shaping our next steps in the systematic development process. First, we had planned to fully develop the guided self-help component of the prototype, but this was logistically challenging especially that this project overlapped with the active COVID-19 pandemic, thus hampering fieldwork activities. This meant that the nurses received only minimal training and were not supervised. Properly developed, this component

will draw-on Ghana's Community-Based Health Planning and Services programme (CHPS) model and have nurses matched with mothers within their communities to provide regular check-ins and problem-solving through their routine home visits. The goal is to bring preventative community-based mental health promotion interventions to the doorsteps of those who need it most through the very successful CHPS anchored on technology. Second, timeliness of the follow-up qualitative interviews could have been improved. We had planned to conduct qualitative interviews within 24 hours of completing every session to minimize risk of recall bias but due to logistical challenges we completed these three months after the delivery of the last intervention session. Reassuringly, the fact that mothers were able to recall salient aspects of the *Obaatanpa* intervention even three months after follow-up suggests that it may have an enduring effect. Third, the relatively low literacy status of the mothers in the formative workshops could have impacted understanding of the complex terms on psychosocial wellbeing and the HAP strategies. We explained these terms in simple language, but we recognize this may not have been sufficient and we did not evaluate this through cognitive interviewing. Fourth, access to a mobile phone was an essential factor for the delivery of *Obaatanpa.* We attempted to establish this during eligibility screening and recruitment for the field-testing; however, it became evident that some mothers had to rely on their spouses' phones. This reliance may have limited their ability to participate fully and could have influenced the quality and consistency of the feedback provided.

## Conclusion

The PREPWELL programme of work has developed a methodology for adapting a treatment-focussed psychosocial intervention for promoting mental health and wellbeing. The design process has resulted in a prototype intervention whose content is determined by the local context, and which builds on best practice. The core content of the intervention consists of psychoeducation, behaviour activation, and problem-solving, carefully woven together to target six domains of psychosocial wellbeing. The data from this study suggest that comprehension and application of HAP concepts were most salient to participants. Problem-solving and shared learning through stories and advice also resonated. However, pleasurable activities – though possibly embedded – were less explicitly recalled, indicating an area for possible emphasis or clearer articulation in future iterations. The development process is continuing into phase two of the PREPWELL programme which will involve a large pilot study to further refine *Obaatanpa* and evaluate implementation outcomes. Given the central role of social determinants of mental health and the proven value of social groups, *Obaatanpa* could be strengthened through the addition of a dedicated module that facilitates linkages to community self-help groups. Ultimately, the PREPWELL programme is designed for integration into Ghana's comprehensive Reproductive, Maternal, Newborn, Child and Adolescent Health and Nutrition programme.

## Supporting information

**S1 Text. Guide.**
(PDF)

## Acknowledgments

The role of Viamo (Swebatu Amadu Salifu) in designing and hosting the digital intervention is acknowledged with gratitude. The authors express their sincere appreciation to all the participants who played a crucial role by providing data for this study.

## Author contributions

**Conceptualization:** Benedict Weobong.

**Data curation:** Benedict Weobong, Solomon Nyame, Kenneth Ae-Ngibise.

**Formal analysis:** Benedict Weobong, Solomon Nyame, Dzifa Attah, Kenneth Ae-Ngibise.

**Funding acquisition:** Benedict Weobong.

**Investigation:** Benedict Weobong.

**Methodology:** Benedict Weobong, Dzifa Attah, Betty Kirkwood, Angela Ofori-Atta, Kwaku Poku Asante, Philip Baba Adongo.

**Project administration:** Benedict Weobong, Solomon Nyame, Kwaku Poku Asante.

**Resources:** Benedict Weobong.

**Supervision:** Benedict Weobong, Kenneth Ae-Ngibise, Joseph Osafo, Betty Kirkwood, Angela Ofori-Atta, Kwaku Poku Asante, Philip Baba Adongo.

**Writing – original draft:** Benedict Weobong.

**Writing – review & editing:** Solomon Nyame, Dzifa Attah, Kenneth Ae-Ngibise, Joseph Osafo, Betty Kirkwood, Angela Ofori-Atta, Kwaku Poku Asante, Philip Baba Adongo.

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
