## [Decision Letter · Decision Letter 0]

2 Nov 2025

PMEN-D-25-00438

Programme for the Effective Promotion of Maternal Psychosocial Wellbeing (PREPWELL) in Ghana: Development and Field-testing of a mHealth Intervention in a Rural Setting.

PLOS Mental Health

Dear Dr. Weobong,

Thank you for submitting your manuscript to PLOS Mental Health. After careful consideration, we feel that it has merit but does not fully meet PLOS Mental Health’s publication criteria as it currently stands. Therefore, we invite you to submit a revised version of the manuscript that addresses the points raised during the review process.

We look forward to receiving your revised manuscript.

Kind regards,

Lambert Zixin Li, Ph.D.

Academic Editor

PLOS Mental Health

Journal Requirements:

1. Please ensure that your Ethics Statement is available in its entirety at the beginning of your Methods section, under a subheading 'Ethics Statement'.

2. Please upload separate figure files in .tif or .eps format. Also, remove the figures from your manuscript file but keep the legends.

https://journals.plos.org/mentalhealth/s/figures

https://journals.plos.org/mentalhealth/s/figures#loc-file-requirements

3. In the online submission form, you indicated that “The data underlying the results presented in the study are available from the University of Ghana data repository. The data will be made accessible and will be availed upon reasonable request, by completing an ‘Expression of Interest form’.”.

3. Uploaded as supplementary information.

Additional Editor Comments (if provided):

Dear Authors,

Thank you for submitting your manuscript to PLOS Mental Health. The paper addresses an interesting and important topic, focusing on an understudied population with potential implications for clinical practice.

Reviewers with relevant expertise found the study valuable but raised significant concerns regarding its scientific validity and overall rigor. Substantial revision is needed before the manuscript can be reconsidered. Please revise thoroughly in response to the reviewers’ feedback and provide a detailed response-to-reviewers memo outlining how each point has been addressed.

We look forward to receiving your revised submission.

Kind regards,

Lambert Zixin Li, PhD

Reviewers' comments:

Reviewer's Responses to Questions

**Comments to the Author**

1. Does this manuscript meet PLOS Mental Health’s publication criteria? Is the manuscript technically sound, and do the data support the conclusions? The manuscript must describe methodologically and ethically rigorous research with conclusions that are appropriately drawn based on the data presented.

Reviewer #1: Yes

Reviewer #2: No

Reviewer #3: Yes

Reviewer #4: Partly

2. Has the statistical analysis been performed appropriately and rigorously?

Reviewer #1: Yes

Reviewer #2: No

Reviewer #3: Yes

Reviewer #4: No

3. Have the authors made all data underlying the findings in their manuscript fully available (please refer to the Data Availability Statement at the start of the manuscript PDF file)?

Reviewer #1: Yes

Reviewer #2: Yes

Reviewer #3: Yes

Reviewer #4: No

4. Is the manuscript presented in an intelligible fashion and written in standard English?

Reviewer #1: Yes

Reviewer #2: Yes

Reviewer #3: Yes

Reviewer #4: No

5. Review Comments to the Author

Reviewer #1: Programme for the Effective Promotion of Maternal Psychosocial Wellbeing

(PREPWELL) in Ghana: Development and Field-testing of a mHealth Intervention in a

Rural Setting.

The abstract and introduction sounds very robust, clear with strong aims and evidence on arguments on how the approaches to publications, research ethics and publication ethics.

I like the use of bar charts and graphs to analyse the mental health conditions of importance to participants in figures 2, 3, 4.

Makes more visually appealing to readers and easier information to digest and more engaging to analyse results.

Table 2: Evolving Obaatanpa Intervention Framework: Application of core HAP strategies to promote psychosocial wellbeing

Good evidence of research to promote psychosocial well being.

I like the structure of the paper and the bold titles and headings to separate the paragraphs and provide concrete substantial evidence of research and data analysis of maternal mental health services for mothers for example, good thorough information and I like the statistics and tables shown to represent the numbers who utilise mental health services.

Improvement of suggestions

You could perhaps include some personal case studies or anecdotes from participants who suffered from specific mental health conditions or mothers who have used maternal health services to flesh out the Paper and make it more engaging and interesting to read and adds depth to the analysis of research.

You can try to see if you can interview some participants from the study within recent years of service from 2020-2025 to identify who has suffered these conditions and used those services to improve pshycho social wellbeing.

Over all it is a fascinating read with some great research, learnt lots from this study, really a eye opener to have known how much mental health needs are met in Ghana and the mental health intervention services and programs help mothers and families out. Especially learning about Obaatanpa Intervention Framework and its HAP strategies to help advocate psycho social well being and to improve the mental health of many participants.

I would double check spelling, grammar and sentence structure, make sure all citations and references are dated correctly, in alphabetical order according to your style guide and make sentences clear and to point, try to avoid paraphrasing and jargon, adding in useful quotes and include relevant (present day) facts/stastics with your arguments validates your points more especially in the middle of your paper.

Also expand on your conclusion, The development process is

continuing into phase two of the PREPWELL programme which will involve a large pilot

study to further refine Obaatanpa and evaluate implementation outcomes.

Where do you see the improvements of the programme in Mental Health, in five years time in the future in terms of Ghana's social and mental well being, what specific services will you implement for revolutionising the mental health system for the benefits for participants for example providing yoga/meditation/ spiritual reiki healing sessions for healing mental well being/ empowering positive mental health or providing spiritual retreats for mothers and children or families or providing social community engaging courses/workshops/facilities to help empower community engagement, making friends with communities and providing sports/exercise/music or art classes, to help improve their mental wellbeing for the positive and for balancing mind, body and spirit, to understand mental health more and how mental health needs can be fully in met in providing non judemental kind, empathetic and compassionate, councillors or psychoanalysts or reiki healers/guiders to help with stress, anxiety, psychological conditions such as psychosis, depression, schizophrenia, bipolar regardless of class, age, nationality, identity/heritage, nomadic groups/communities, background, colour, gender, religion/faith, belief to be very fair.

To be very honest just needs minor amendments from above suggestions to be very fair, otherwise good to get published.

Well done.

All the best

:)

Reviewer #2: Thank you for the great initiative to address mental health issues for this population of women in Ghana who are often underrepresented and whose well-being might be neglected. Reading the manuscripts and the limited resources during the pandemic, I can't imagine the amount of effort devoted to succeeding in this meaningful project. Definitely worth reading; here are my suggestions, primarily aimed at enhancing and advancing the manuscript.

1.Style and Format: Kindly ensure the manuscript is submitted as per the suggestions by PLOS Mental Health and the link, particularly:-

1.1.Reference style: change to PLOS uses “Vancouver” style, as outlined

1.2.Abbreviations: some of the abbreviations seem to be missing in the first appearance in the text, refraining from using non-standard abbreviations unless they appear more than three times in the text and keep it minimum

1.3.Result reporting style: Kindly adhere to the format

1.4.Reference list: Please update based on guidelines

2.Introduction

2.1.Page 3, Line 50 -53: Rephrase to academic writing style, e.g. Prevalence of perinatal depression in LMICs is xx% antenatally and xx% postnatally, respectively.

2.2.Page 3, Line 59: Please specify the type of treatment

2.3.Page 4, Line 81-84: Please rephrase, having limitations to understand.

2.4.Page 5, Line 96-97: Please follow the reference and citation guidelines

2.5. Please include the brief demographic characteristics and the issues of Ghana, such as low gadget ownership and data subscription, infrastructure challenges, attrition rate in treatment, and any existing and former digital MH studies conducted in Ghana.

2.6.Please relook into the Introduction and the sequence. Reading the introduction, readers may have limitations in understanding how this app converges or diverges with existing digital MH platforms or the research gap.

3.Method & Material

3.1.Page 6, Line 138, Please provide keywords of literature reviews

3.2.Page 8, Line 167-190: Please provide references

3.3.Page 8, Line 184: Please rephrase. Could you help the reader to understand what “ranking (12) and visualisation…” means?

3.4.Page 8, Line 187-199: Please summarise, consider including the questions in supplementary material.

3.5. Page 8. 202-204 Please rephrase. Could you help the reader understand what DA and BW are? Not a common abbreviation.

3.6.Page 10, Line 213-221: Please specify and explain the translation process, if the research employed any linguistic experts or back-to-back translation, etc.

3.7.Page 11: Please specify the participants' recruitment inclusion and exclusion criteria and how the PHQ was administered and why the PHQ was administered, as well as the analysis. PHQ has a cut-off score and provides a brief introduction of PHQ under the instrument sub-section. Are your studies looking for patients with depressive symptoms?

3.8.Page 12: Please provide a sample size calculation or power analysis.

3.9.Please re-examine the flowchart, suggesting drafting a flowchart with CONSORT

4.Result

4.1.Page 14, Line 286-299: Please focus on result reporting.

4.2.Page 15-17: Please rephrase and follow the result reporting guideline.

4.3. Page 18/ Table 2: Please check the reporting guideline and relook into the title of the table and table formatting. Please provide the phrase and the number of sessions for each segment in your study.

4.4.Please check the page number and the line number; all are missing. Please rephrase and follow the result reporting guideline.

4.5.The PHQ result was mentioned without a table, and the comparison result. Is PHQ scoring part of the measurement of the effectiveness of the study?

4.6.Please relook into the mental health digital app development stage/ study design, the intervention framework shall follow a structured & clear study design that spans several phases. Some of the phrases are achieving different goals, and the development process is unclear. Upon reviewing your manuscript, I suggest breaking it into 5-7 Phrases. - Phase (1) Need assessment with mixed method, e.g. Stage A … Stage B…; Phase (2) Baseline; Phase (3) Programme development; Phase (4)…; Phase (5) Implementation of trials; Phase (6) Follow-up and evaluation.

4.7.Please consider adapting the CONSORT-style flowchart to visualise the study design, including the number of sessions and outcome/ goals.

4.8.Please consider providing a table for Fig. 2-4 with frequency and descriptive data, instead of a graph. Fig. 5 is missing, or the file is unreadable.

5.Discussion

5.1.Please rephrase in an academic writing style.

5.2.Please help the readers understand why gadget ownership was discussed, but the result section was not highlighted.

5.3.Please help the readers understand why help-seeking behaviours or self-care behaviours changed after the intervention, and based on your findings.

5.4.Please provide references to support your findings and split them into a few paragraphs based on your findings, e.g., the clean interface of the apps and the narrative-based intervention.

5.5.Please explain the effectiveness of the apps, for example, whether the content helped the participants who had low media literacy easily grasp the self-help tips. Please provide a reference. Please help the readers understand what “promising result” is.

5.6.Please explain how the apps link with positive psychology? If positive psychology is part of the theoretical framework, please include it in the study design.

Reviewer #3: The manuscript demonstrates strong methodological and ethical rigor, offering a valuable contribution to global mental health research. Minor revisions are advised to improve language consistency and overall clarity.

Reviewer #4: Introduction

1)The abbreviation BA is not written in its full form. Please write the full form of BA intervention in the introduction to clearly state its purpose. Also a little detail can be added in the introduction section about BA interventions explaining what they are.

2)In the introduction section the BA intervention is linked to promoting psychological wellbeing but the population is not specified. Is it the pregnant women of the general population that BA interventions had shown improvements with. Perhaps the idea of BA interventions linking to maternal psychological wellbeing should be made clear.

3)The introduction section does not state what mental health problems are present among women who are pregnant. And also, the current intervention is designed for which mental health problems. (eg depression, anxiety, major depressive disorders etc.)

4)Clearly state if the intervention is a type of BA intervention or something else. (clearly mention that BA is a type of evidence-based psychological treatment which this intervention will follow)

5)Write the full form of HAP in the introduction section where it is first introduced.

6)Clear rationale for why this evidence-based treatment was chosen

7)Use abbreviations correctly throughout the manuscript.

Methodology

1)Inclusion/exclusion criteria is not clearly defined (target population, age group, mental health condition, language, target sample size etc.)

2)Recruitment method not described (community, clinics, etc.)

3)The manuscript mentions that a rapid review was conducted to gather existing evidence on the topic; however, the review itself is not cited, published, or made accessible. For transparency and reproducibility, please clarify: 1) Whether the review followed a protocol (and if so, provide registration details or an appendix). 2) Whether the review results are available (e.g., as supplementary material, institutional report, or online dataset). 3) The databases searched, inclusion/exclusion criteria, and search terms.

4)The introduction for HAP should be moved to introduction section and made clear with the objectives of the study.

5)There is no step C?

6)Mention the steps in the stages in the paragraph. Eg, the first stage involves 3 steps: (A), (B), (C) and then start describing the steps to make things clear.

7)Use proper and similar referencing in the manuscript.

8)The manuscript mentions the use of “structured workshop guides” to conduct participatory learning and action sessions. However, the source and development process of these guides are not described. For transparency and replicability, please specify:

Whether the guides were adapted from existing manuals, developed, or co-created with participants.

How the content was structured (number of sessions, key topics, activities).

Whether the materials are accessible (e.g., in supplementary materials or upon request).

9)The manuscript describes the use of participatory approaches (e.g., PLA techniques), but there is no mention of patient or public involvement (PPI) in the design, conduct, or interpretation of the study.

Since participatory research inherently values collaboration with community members, it would strengthen the manuscript to clarify:

Whether and how target participants or community representatives were involved in shaping the study design or materials, and

If not, to include a formal statement on the absence of PPI in accordance with journal guidelines.

10)Citation for the original PHQ-9 paper (Kroenke et al., 2001)? Did you use PHQ-9 as an outcome measure (to assess mental health before and after the intervention)? Was it self-completed or interviewer-administered? How were the incomplete responses handled?

11)The manuscript reports on an intervention but does not specify the study design (e.g., randomized controlled trial, quasi-experimental, pre–post, or pilot). Clearly identifying the research design is essential to understand the level of evidence, sampling method, and appropriate analysis. Please clarify the study design and describe any comparison or control groups, randomization procedures, or rationale for the chosen design.

12)The manuscript does not describe the recruitment process, including how participants were identified, approached, and enrolled, nor does it report how many were eligible versus how many participated. Providing this information, including inclusion/exclusion criteria, recruitment methods, and a flow diagram, is essential for transparency, assessment of potential selection bias, and reproducibility. (a CONSORT-style flow diagram)

13)The information regarding the intervention was not sufficient (time and duration, place, if participants were compensated etc)

14)The Methods section requires significant improvement for clarity and structure. Currently, the sequence of information is not well aligned — details about the intervention, data collection, and analysis appear intermingled, and the overall flow does not follow a logical order. I recommend restructuring the Methods section into clear subsections (e.g., Study Design, Setting and Participants, Intervention, Data Collection, Data Analysis, and Ethical Considerations). Each step should be described sequentially and in enough detail to allow replication.

Results

1)The study reports that 59 participants were enrolled, but no explanation is provided regarding how this number was determined. Please clarify whether this was based on a formal sample size calculation, recruitment feasibility, or convenience sampling. Additionally, provide context regarding eligibility, recruitment, and attrition to help readers interpret the adequacy and representativeness of the sample.

2)The stage 1 says ‘The technology was evaluated to assess the platform’s effectiveness in delivering messages along with user response across sites in five districts from 2011 to 2014.’ But the study time period was mentioned between 2nd September 2022 and 27th October 2022. Which time period is the correct one?

3)If the intervention was to provide mental health services for mothers why were mental health assessment mot conducted?

4) What statistical analysis method was used for result analysis?

5) The data from the analysis is not reported sufficiently.

6)Information regarding the data analysis and the method used should be clearly mentioned in the method section under statistical analysis section which was missing.

7)The number of participants is not consistent throughout the result section and it is confusing to interpret how many participants were involved in the intervention and how many dropped out. (in the starting it mention there are 8 pregnant women and 8 recently delivered but in the later sections it mentions ‘8 out of 13 mothers ‘.)

Discussion

1)The Discussion section mentions that the study co-created a prototype psychosocial wellbeing promotion mHealth intervention. However, the Methods section does not describe how this co-creation process was conducted — including who participated, the steps taken, or how findings informed the prototype. As co-creation is a key outcome and methodological process, it should be explicitly described in the Methods, with sufficient detail to ensure transparency and reproducibility.

2)The section had some point not mentioned before (eg story telling technique was employed). This should be included in the method and result section with data mentioned from the participants.

3)The data was only given in graphical form and should also be given in the numerical form for transparency.

Final Evaluation

despite the relevance and potential impact of the study, the manuscript suffers from major methodological and reporting shortcomings that compromise its transparency, reproducibility, and scientific rigor. The paper requires significant revision and clarification before it can be considered for publication.

Clearly identify the study design and align the methodology, data collection, and analysis accordingly.

Expand the Methods to include recruitment, eligibility criteria, sample size rationale, and detailed procedural steps. A CONSORT-style flow diagram would strengthen reporting transparency.

Describe the analytical process in sufficient detail, including data preparation, coding, thematic development, statistical tests (if applicable), and software used.

Include a section describing how participants or community members were involved in co-design or decision-making processes.

Provide complete numerical data in tables, including descriptive and inferential statistics. Graphs should complement—not replace—numeric reporting.

Move or expand this content into the Methods section and describe the co-creation process systematically.

6. PLOS authors have the option to publish the peer review history of their article (what does this mean?). If published, this will include your full peer review and any attached files.

**Do you want your identity to be public for this peer review?** For information about this choice, including consent withdrawal, please see our Privacy Policy.

Reviewer #1: **Yes:** Miss Neelam Jayendra Shah

Reviewer #2: No

Reviewer #3: **Yes:** Charles Ganaprakasam

Reviewer #4: No

Figure Resubmissions:

---

## [Decision Letter · Decision Letter 1]

26 Jan 2026

PMEN-D-25-00438R1

Programme for the Effective Promotion of Maternal Psychosocial Wellbeing (PREPWELL) in Ghana: Development and Field-testing of a mHealth Intervention in a Rural Setting.

PLOS Mental Health

Dear Dr. Weobong,

Thank you for submitting your manuscript to PLOS Mental Health. After careful consideration, we feel that it has merit but does not fully meet PLOS Mental Health’s publication criteria as it currently stands. Therefore, we invite you to submit a revised version of the manuscript that addresses the points raised during the review process.

We look forward to receiving your revised manuscript.

Kind regards,

Lambert Zixin Li, Ph.D.

Academic Editor

PLOS Mental Health

Journal Requirements:

Additional Editor Comments (if provided):

Dear Authors,

Thank you for submitting your revised manuscript to PLOS Mental Health. The reviewers appreciate the improvements made in response to the initial comments. A small number of issues remain that should be addressed before we can proceed further.

Please revise the manuscript to carefully respond to the remaining reviewer comments and clarify any points that still require refinement.

We look forward to receiving your updated manuscript.

Sincerely,

Lambert Zixin Li, PhD

Reviewers' comments:

Reviewer's Responses to Questions

**Comments to the Author**

1. If the authors have adequately addressed your comments raised in a previous round of review and you feel that this manuscript is now acceptable for publication, you may indicate that here to bypass the “Comments to the Author” section, enter your conflict of interest statement in the “Confidential to Editor” section, and submit your "Accept" recommendation.

Reviewer #1: All comments have been addressed

Reviewer #3: All comments have been addressed

2. Does this manuscript meet PLOS Mental Health’s publication criteria? Is the manuscript technically sound, and do the data support the conclusions? The manuscript must describe methodologically and ethically rigorous research with conclusions that are appropriately drawn based on the data presented.

Reviewer #1: Yes

Reviewer #3: Yes

3. Has the statistical analysis been performed appropriately and rigorously?

Reviewer #1: Yes

Reviewer #3: Yes

4. Have the authors made all data underlying the findings in their manuscript fully available (please refer to the Data Availability Statement at the start of the manuscript PDF file)?

Reviewer #1: Yes

Reviewer #3: Yes

5. Is the manuscript presented in an intelligible fashion and written in standard English?

Reviewer #1: Yes

Reviewer #3: Yes

6. Review Comments to the Author

Reviewer #1: So far all the comments and feedback by my first peer review comments have been addressed and met according to the criteria and guidelines.

So far I do think the research ethics is very strong and adds rigour to the paper, in terms of publication ethics the paper meets the answers and needs of specific question to the arguments, the Programme for the Effective Promotion of Maternal Psychosocial Wellbeing (PREPWELL) in Ghana: Development and Field-testing of a mHealth Intervention in a Rural Setting.

I also think the referencing and citing has been meet with fully accuracy according to the style guide of referencing and use of tables/figures to analyse the questions and arguments addressing the methods of mental health interventions in Ghana in promotion of psycho social well being, also using technology and other modern technological pathways to find solutions to problems to tackle the interventions of mental health of women and men in Ghana in terms of providing a better life style in their rural setting and help positively empower and make a difference to their psychosocial well being during maternity.

The conclusion is strong and conclusive summarising the main key points of paper.

It is a very fascinating subject and to learn about other countries development of Mental Health Interventions in rural settings to help psychosocial well being is really a eye opener, learnt lots, really great in depth research and I do like the use of case studies.

My only little concerns/last edits to the paper to make it seem outstanding.

is maybe adding some more anecdotes or personal experiences or accounts of women in Ghana who have been through maternity in rural settings facing through psycho social mental health problems and how mental health interventions can help them and monitor/assesses their mental health needs in terms of digital technology guiding them in rural areas too in helping them farming or better improve their life styles, through getting some interviews or case studies of personal video questionnaires just to flesh out the paper a bit more, and make it flourish with some qualitative perspectives and humanising the paper, may invite more readership to connect to the paper too.

i.e. how their personal experiences, how are they coping with those conditions/ specific symptoms, is there enough done to ensure the mental health interventions can help them in Ghana with psychosocial well being and mental well being in helping them attend more skills learning classes, community grass roots maternity focus groups with other mothers who are going through similar experiences, so all the mothers are not left isolated or alone in rural communities, to help attend group social activities to make and meet new friends/like minded people, in rural settings for all mothers too, i.e. supportive of rural businesses together, cooking, art or textiles, sports/yoga/reiki/meditation, music/dance, or fashion or teaching new skills i.e reading books, learning computers/PC etc.

So they can improve their mental well being together.

You can include in conclusion too, perhaps if you want the next future steps on how the Mental Interventions will help progress positivity in mental well being in Ghana.

And forwardness in encouraging Ghana's rural public to help psycho social well being especially in Maternity to help empower their mind, body spirit by introducing yoga/reiki/meditation to be very fair to keep the positive mental health for future years to come, e.g. in next 5 to 10 years Ghana's maternal program of development is progressed by these mental health interventions and helped specifically the mother's psychosocial well being by these solutions i.e. mentioned grass roots projects, opening up community classes, social activities, more digital technology in medicine advanced to help aid and measure the mental health of mothers in Ghana, using perhaps new inventions or interventions like mental health monitors of maternal patients in pharmacies/ hospitals etc. to assess and measure the mental health needs of mothers more accurately, which in rural areas can be invented and distributed free in those communities, to assess happy/sad/anxiety/mood disorders/social/emotional levels of mental health too for example.

I like the amendments considered and changes/feedback according to each peer reviewer, having thoroughly read the paper, it is good to get published with just a minor re edits if you can consider my feedback above.

Very well done for the paper.

I wish you all the very best of luck for its publication.

Reviewer #3: Claims related to scalability and broader impact in the discussion should be framed more cautiously to align with the small sample size and exploratory nature of the field-testing phase. With minor revisions to strengthen interpretive balance, the manuscript is suitable for publication.

7. PLOS authors have the option to publish the peer review history of their article (what does this mean?). If published, this will include your full peer review and any attached files.

**Do you want your identity to be public for this peer review?** For information about this choice, including consent withdrawal, please see our Privacy Policy.

Reviewer #1: **Yes:** Miss Neelam Jayendra Shah

Reviewer #3: No

Figure Resubmissions:

---

## [Decision Letter · Decision Letter 2]

18 Feb 2026

Programme for the Effective Promotion of Maternal Psychosocial Wellbeing (PREPWELL) in Ghana: Development and Field-testing of a mHealth Intervention in a Rural Setting.

PMEN-D-25-00438R2

Dear Dr. Weobong,

We are pleased to inform you that your manuscript 'Programme for the Effective Promotion of Maternal Psychosocial Wellbeing (PREPWELL) in Ghana: Development and Field-testing of a mHealth Intervention in a Rural Setting.' has been provisionally accepted for publication in PLOS Mental Health.

Best regards,

Lambert Zixin Li, Ph.D.

Academic Editor

PLOS Mental Health

Dear Authors,

We are pleased to inform you that your manuscript has been accepted for publication. Thank you for your thoughtful revisions.

Best regards,

Lambert Zixin Li, PhD

Reviewer Comments (if any, and for reference):

Reviewer's Responses to Questions

**Comments to the Author**

1. If the authors have adequately addressed your comments raised in a previous round of review and you feel that this manuscript is now acceptable for publication, you may indicate that here to bypass the “Comments to the Author” section, enter your conflict of interest statement in the “Confidential to Editor” section, and submit your "Accept" recommendation.

Reviewer #1: All comments have been addressed

Reviewer #3: All comments have been addressed

2. Does this manuscript meet PLOS Mental Health’s publication criteria? Is the manuscript technically sound, and do the data support the conclusions? The manuscript must describe methodologically and ethically rigorous research with conclusions that are appropriately drawn based on the data presented.

Reviewer #1: Yes

Reviewer #3: Yes

3. Has the statistical analysis been performed appropriately and rigorously?

Reviewer #1: Yes

Reviewer #3: Yes

4. Have the authors made all data underlying the findings in their manuscript fully available (please refer to the Data Availability Statement at the start of the manuscript PDF file)?

Reviewer #1: Yes

Reviewer #3: Yes

5. Is the manuscript presented in an intelligible fashion and written in standard English?

Reviewer #1: Yes

Reviewer #3: Yes

6. Review Comments to the Author

Reviewer #1: Thank you for adhering to all the comments and suggestions for the questions above regarding research ethics, publication ethics and in general your paper.

I agree with all your amendments and improvements, thank you for clarifying my feedback and comments and discussions.

I understand fully this is at a evaluation stage so case studies can't be given just yet or accessed to any testimonials. Thank you for clarifying your paper and discussion into the next steps.

Very organised and efficient.

Obaatanpa, seems like a fantastic mini genius phone device from prototype to help advance technology in improving inclusivity/equity from mothers from low SES who may only be able to afford such feature phones.

Thank you for highlighting the prototype and that you are going to be collaborating with the Ghana Health Service to ensure the potential uptake of Obaatanpa into Ghana’s comprehensive Reproductive, Maternal, Newborn, Child and Adolescent Health and Nutrition programme – similar to The Mobile Technology for Community Health in Ghana (MOTECH), that is great news because they can help with the technology to advance and give mental aid and support to mothers there which may not have access to mobile technology or knowing how to use it just yet, and to give that teaching of nutrition can help too, with good nutrition and expert health advice that would help improve their lives for better healthier lifestyle choices too in terms of changing diet/exercise routines/inclusive of sports etc .

So far very brilliant responses. Hope all in future will be a big success and the prototype of the Obaatanpa phone devices (advanced technology) can make a positive impact on the mother's mental health in terms of social, emotional, mental and overall physical well being and can make a positive lifestyle difference to their Motherhood lives, especially coming from poverty. To create positive innovative changes in the Ghana's community maternal health future.

It will be exciting to measure the impacts of mental health and low SES from these devices to investigate the results in a survey/questionnaire analysis with Infographics/charts with clear evaluation and thorough data analysis and compare trends from this year to last year for example to indicate any levels of changes/improvements of using the device how has the mothers in Ghana community mental health improved since using the Obaatanpa phone devices by using line/bar graphs or pie charts from a decade from now, in next 10 years 2026 to 2036? for example compared to no use of technology and how that has impacted their mental health with or without any phones/technological devices etc. Therefore create a end report/research paper and summarise the findings in a evaluation research paper/report worldwide to raise awareness of the subject to the public.

The research ethics utilised is very robust, ethical, authentic and credible with up to date relevant data and sources.

Very well done, for me as a reviewer, it is now at a publishing stage due to all the feedback and suggestions of improvements have been thoroughly taken on board and fully checked with grammar, spelling, and puntuanction and references/citations double checked.

All the best with the Research programme and the publication of this Research Paper.

Reviewer #3: Overall, this manuscript makes a meaningful contribution to the global mental health literature, particularly in relation to the adaptation of technology-based psychosocial interventions in resource-limited contexts, and is suitable for publication.

7. PLOS authors have the option to publish the peer review history of their article (what does this mean?). If published, this will include your full peer review and any attached files.

**Do you want your identity to be public for this peer review?** For information about this choice, including consent withdrawal, please see our Privacy Policy.

Reviewer #1: **Yes:** Miss Neelam Jayendra Shah

Reviewer #3: **Yes:** Charles Ganaprakasam
